# Vaccination Strategies in a Potential Use of the Vaccine against Bovine Tuberculosis in Infected Herds

**DOI:** 10.3390/ani12233377

**Published:** 2022-12-01

**Authors:** Feliciano Milián-Suazo, Sara González-Ruiz, Yesenia Guadalupe Contreras-Magallanes, Susana Lucía Sosa-Gallegos, Isabel Bárcenas-Reyes, Germinal Jorgé Cantó-Alarcón, Elba Rodríguez-Hernández

**Affiliations:** 1Facultad de Ciencias Naturales, Universidad Autónoma de Querétaro, Santiago de Querétaro 76010, Mexico; 2Centro Nacional de Investigación Disciplinaria en Fisiología y Mejoramiento Animal, Instituto Nacional de Investigaciones Forestales, Agrícolas y Pecuarias, Ajuchitlán 76280, Mexico

**Keywords:** *Mycobacterium bovis*, bTB vaccines, cattle tuberculosis, vaccine strategies

## Abstract

**Simple Summary:**

Bovine tuberculosis is a chronic infectious disease caused by *Mycobacterium bovis* characterized by the formation of tubercles in any organ or tissue. This disease affects various animal species and humans; therefore, it represents a significant veterinary and public health problem. Bovine tuberculosis control is based on the tuberculin test and the disposal of reactor animals. In underdeveloped countries, this strategy has not been successful due to the lack of economic resources to compensate producers for their slaughtered animals. Recently, it has been suggested to incorporate the Bacillus Calmette–Guérin (BCG) vaccine to reduce the disease prevalence by reducing the number of animals with lesions, the number of lesions per animal and bacterial load. Several experiments have shown the efficacy of different vaccine formulas against bovine tuberculosis. In Mexico, there is a growing interest, both from producers and authorities, to integrate the vaccine into current control programs, especially in dairy cattle, where the prevalence of the disease is high, and overcrowding favors the dissemination.

**Abstract:**

Bovine tuberculosis (bTB) is a disease of cattle that represents a risk to public health and causes severe economic losses to the livestock industry. Recently, one of the strategies recommended for reducing the prevalence of the disease in animals is the use of the BCG vaccine, alone or in combination with proteins. It has been shown that the vaccine elicits a strong immune response, downsizes the number of animals with visible lesions, and reduces the rate of infection as well as the bacillary count. This paper, based on scientific evidence, makes suggestions about some practical vaccination alternatives that can be used in infected herds to reduce bTB prevalence, considering BCG strains, vaccine doses, routes of application, and age of the animals. Our conclusion is that vaccination is a promising alternative to be included in current control programs in underdeveloped countries to reduce the disease burden.

## 1. Introduction

Bovine tuberculosis (bTB) is a chronic infectious disease characterized by the formation of tubercles, lesions with pus, generally semi-calcified and crackling when cut, which affects various animal species and humans [1,2]. In addition to the economic losses due to low production, premature disposal, confiscations at the slaughterhouse, and limitations in marketing, the estimated global economic loss is about USD 3 billion per year with more than 50 million cattle infected [3]; in England alone, the cost of TB is about GBP 120 million per year [4]. In Mexico, from 2015 to 2017, economic losses due to the death and slaughter of cattle because of TB was estimated at USD 5.5 per head per year [5]; the cost of the TB control program from 2010 to 2020 in this country was about USD 120,340 [6]. TB is a risk to public health; people can become infected by close contact with sick animals or by the consumption of raw milk [7,8,9].

bTB has a worldwide distribution with variable prevalence rates by region and by country, with rates as high as 50% in some regions of the African continent [10,11,12,13], and for the European continent, the overall proportion of cattle herds infected has remained very low (0.9%) [14,15,16]. In the north of the American continent, in the United States of America and Canada, the prevalence is extremely low, concentrated in some regions where it is shared between livestock and wildlife [17,18]. In Central and South America, the disease is endemic, with higher prevalence in populations of dairy cattle [19,20]. bTB control is based on the tuberculin test and the disposal of reactors, in addition to the epidemiological surveillance in slaughterhouses. The main ante mortem tests are the tuberculin skin test (TST) and the interferon-gamma release assay (IGRA-IFN-γ). Both tests detect cell-mediated immune responses. The TST is the primary screening test prescribed by the World Organization for Animal Health (WOAH) for bTB surveillance programs [21]. Tests based on the application of tuberculin have serious limitations due to their low sensitivity and specificity [7,22,23].

Developed countries have been successful in reducing the prevalence of bTB to minimal rates with the “test-and-slaughter” strategy. Affected herds are compensated with almost full payment of the commercial value of the culled animals [24,25]. In the case of underdeveloped countries, this strategy has not been successful due to the lack of economic resources to compensate farmers for their slaughtered animals [26]. Currently, no country in the world uses vaccination to control bovine tuberculosis, despite the fact that the BCG vaccine has been used for approximately 100 years to prevent the disease in children. One of the reasons is that it interferes with the tuberculin test, a test used to identify and eliminate sick animals in herds and establish prevention measures, such as quarantines or depopulation; there are no resources to compensate for the eliminated animals, more evidence of the vaccine is required; or for legal reasons, as is the case in the European Union (EU) [26]. In some countries, there is also concern about restrictions that could be imposed on the movement of vaccinated cattle; vaccinated animals would be positive to the tuberculin test and, therefore, suspected of having the disease. However, in several countries, such as Mexico, there is a growing interest, both from farmers and authorities, in integrating the vaccine into the current control programs, especially in dairy cattle, where the prevalence of the disease is high.

Mass vaccination of livestock must have clear and achievable objectives, at both the herd and regional, or even national, levels, considering the prevalence of the disease and the conditions and purpose of the production unit, for example if the unit needs to trade cattle and their products or need to move animals to free areas. Vaccination in low-prevalence regions, where cattle are moved frequently between regions and for exportation, would not be recommended; under those conditions, partial or total herd depopulation, and looking for a free status, could be a better option. In Mexico, for example, vaccination is not recommended for regions with low prevalence (<0.5%), where “test-and-slaughter” has been successful and the exportation of calves to the United States of America is important. Vaccination is considered for regions where the “test-and-slaughter” is not an option.

In Mexico, vaccination is focused on dairy cattle, where the prevalence of TB is ≈16% [19,20]. The objective is to reduce the prevalence to levels where the strategy of “test-and-slaughter” is economically feasible, for both the farmer and the official program. Currently, dairy farmers participate little in the control program because they do not trust the tuberculin test. In addition, they argue that milk is sent to pasteurization plants, they do not export animals, replacements are costly, and production rates need to be maintained; young animals produce less milk than adults. In this particular case, while reducing the prevalence of TB, vaccination could be an incentive for milk farmers to participate in the national control program; they are not against the program, they are against eliminating animals based on tests with such a low sensitivity such as the tuberculin test. Furthermore, milk producers in Mexico do not want to have bTB, as that is like a stigma among peers.

Vaccination has also the objective of increasing animal welfare, and reducing economic losses by preventing animal death, carcass disposal, low production, and restrictions on the sale of animals. More important than that, it reduces the risk of transmission to people; in humans, *M. bovis* can cause a disease similar to that observed with *M. tuberculosis* [7]. Vaccination also has the objective of preventing the transmission of TB to wildlife, to protect endangered animals and avoid reservoirs in those populations [26].

## 2. Review

### 2.1. Experimental Studies with BCG

Calmette and Guérin in 1911 [27] first reported their work on BCG vaccination in cattle, showing that relatively high doses of BCG (20 mg) could induce protection against an experimental challenge with *M*. *bovis*. Contemporary trials of vaccination against bTB began in the 1990s, with greater intensity from the year 2000 on, where the main objective was to determine its efficacy, both experimentally—testing with different BCG strains, vaccines, and challenge inoculation and doses—and in field trials—in domestic and wild animals [28,29,30]. These works included species such as cattle, goats, sheep, feral pigs, white-tailed deer, red deer, possums [31,32,33,34,35,36,37], and badgers [38,39], where different levels of protection were observed.

Seen globally, one of the complications when drawing conclusions from the large number of experiments about the efficacy of the BCG to attack bTB is the diversity of conditions in which the experiments are carried out, with most of them having small sample sizes due to the high cost of the experimental animals. In addition, there is diversity in the strains of BCG, vaccine formulations, vaccine and challenge doses, routes of application, and age of the experimental animals. However, in spite of all those differences, most of them report promising results in terms of pathological damage [40,41,42,43].

The conclusion of these studies is that, in most cases, the vaccine, alone or in different formulations combined with proteins, is capable of inducing a strong immune response, reducing the number of animals with visible lesions, the quantity and the magnitude of the lesions, as well as the bacterial load and, consequently, it is capable of preventing the spread of the disease. Vaccine efficacy reports range from 0 to 75% based on lesion score, lesion number reduction, lesion size, and bacillary load [29,44,45,46] (Table 1). A recent meta-analysis with 24 papers reported an overall vaccine efficacy of 25% in preventing lesions and isolating the pathogen [47]; this meta-analysis concluded that BCG vaccination may help in the control of bTB in endemic settings with no effective bTB control programs.

In determining vaccine efficacy, it has been observed that low doses are better than high doses [27], that it is better if applied to animals at birth [51], and that the use of a booster with proteins improves protection [46,52,53]. In a field study, it was observed that the vaccine induces an immune response similar to that observed experimentally, and that it is totally safe in pregnant animals, harmless to the mother or the fetus [49]. In addition, it has been shown that vaccination with BCG and/or Purified Protein Filtrate (CFP) does not induce an IFN-gamma response with low-molecular-weight antigens such as CFP-10 and ESAT-6, which allows differential diagnosis [54,55,56]. Short-term BCG revaccination has been reported to have a negative effect on vaccine efficacy. Animals vaccinated within the first hours of life or at 6 weeks had similar efficacy, contrary to what happened with those vaccinated at 8 h and revaccinated at 6 weeks, where protection was lower [29]. Thus, there is agreement among the authors that the BCG vaccine, alone or in combination with proteins, would be useful to reduce the incidence, and consequently the prevalence, of bovine tuberculosis in infected herds if it is incorporated into existing control programs [44,46,49,57].

It is worth emphasizing that, although the experimental models try to simulate natural infections, it is clear that the challenge doses used in the experimental studies are much higher than those expected in natural environments [58], which suggests that, in general, the efficacy of the vaccine will be better in the field, especially if the majority of the population at risk is vaccinated, to achieve herd immunity by vaccinating between 70 and 80% of the population. In relation to cost, one study found that producers in the UK would be willing to pay for the use of a vaccine when the price is about GBP 17 per animal per year [59]. For producers in many countries where the prevalence of tuberculosis is high, this price would be unaffordable, so one of two things would need to happen: either the governments would need to establish some kind of subsidy or the price of the vaccine would have to be considerably reduced. According to informal comments of the producers in Mexico, they would be willing to pay a price of between USD 7 and 10 per dose. However, the real cost of the vaccine in an open market remains to be determined. On the one hand, as with other drugs, the price per dose could be lower as production and demand increase, and on the other hand, the cost will be related to the benefits of the vaccine; the role of politicians in the vaccination program would be crucial in deciding what the cost for the producer will be. This role is going to depend on the country’s goals.

### 2.2. Field Evaluation of the Cattle TB Vaccine

There have been many experimental studies where the efficacy of different vaccine formulas against bovine tuberculosis has been evaluated: different BCG strains [46,60,61]; BCG mutants and field strain mutants [53,62,63]; DNA vaccines [64,65]; protein vaccines [57]; and several others [43]. However, few works have been carried out under the natural conditions of challenge [41,43,48,50]. Some of the reasons for the lack of these kinds of studies are that they are expensive, long-term, and complicated; financial support is not easy to find, and producers are not always willing to allow the management of their cattle for research purposes. For example, they argue that milk production goes down when cattle are moved for vaccination or blood sampling.

Field studies have shown levels of vaccine efficacy from 22 to 86% [42,43,48,49], depending on the parameter used to determine efficacy. These parameters range from positivity to an immunological test with the use of specific antigens, to the quantification and classification of lesions, isolation, and bacterial load [42,46,48]. In general, the reported efficacy is low; however, despite the fact that, experimentally, it has been shown that this is significantly improved when a protein-based booster is used—generally a culture protein filtrate (CFP)—in none of them has this booster been used. This suggests that vaccine efficiency can be improved if this strategy is applied. It has also been shown that the vaccine is safe in pregnant heifers, both for the mother and the fetus [65], and that vaccination has positive side effects on productive and reproductive parameters [50]. A field study that had the opportunity to sacrifice the animals showed variable efficacy depending on the time to sacrifice, with 67.4% when they were sacrificed at one year and 77.4% when sacrificed two years after vaccination [66], where it was also perceived that the vaccine decreased the progress of the disease by having fewer animals with lesions and lesions with lower scores, indicating less pathological damage [26]; several studies have shown similar results (Table 2).

One of the critical points in the use of the BCG vaccine, as occurs with most vaccines, is having indicators of protection or vaccine efficacy in vivo. To date, the two indicators of efficacy identified are the INF-γ cytokines [29,31,46] and IL-22 [75]; therefore, in the near future, perhaps the objective in field work will be to identify the levels of these proteins in vaccinated animals to predict protection and pass to second term the tuberculin tests.

Thinking about the massive use of the vaccine to counteract the effect of bovine tuberculosis, some of the questions that arise are: How and when should we vaccinate? What strain and what dose should be used? What is the ideal anatomical site for vaccine application? At what age should vaccination take place? Additionally, should the entire herd be vaccinated even though some animals are already infected? Vaccination strategies for infected herds can be many and varied, depending on prevalence; management conditions; and short-, medium-, and long-term goals. Among others, some of the strategies that can be used are:How and when to vaccinate.

The age of the animals used in the experiments about BCG efficacy is diverse. Some studies mention that vaccination at early ages, a few days after birth, induces better protection against challenge than when young or adult animals are used [29,45,67]; however, studies in older animals have given similar results [26,29,46,49,76]. The argument for using newborn animals is that in this way, primary infection with environmental mycobacteria is avoided before vaccination, which could lead to a lower efficacy of the vaccine. However, some studies report not having detected this effect [50]. The evidence suggests that the age of the animals at the time of vaccination is irrelevant, that the immune response is very similar in both cases, and that BCG in most of the formulations has shown efficacy.

BCG strain to use.

BCG is the result of a series of successive passages of a culture of *M. bovis* for 13 years by Calmette and Guérin at the Pasteur Institute [77] and from there, the BCG went to different laboratories around the world; therefore, different BCG strains variants came out. The variability of passaging conditions was one of the reasons for those differences [78]. If there are differences between the strains, then it is likely that there are also differences in the immune response to the strain used in the vaccine; however, it has been shown in experimental work in cattle that the BCG strain used does not make a significant difference in the immune response, at least for the strains most commonly used (BCG Danish 1331; BCG Pasteur 1173; BCG Glaxo 107; BCG Tokyo 172-1; BCG Russia-Ir; and BCG Brazil) [39] and in the efficacy of the vaccine, contrary to reports of studies with laboratory animals [79]; therefore, any of the BCG strains can be used, and perhaps the recommendation would be to avoid competition with vaccination programs in people. The BCG strains most frequently used experimentally have been the Danish [42,48,49,70,80], the Pasteur [31,40,67,76], the Tokyo [41,81], and the Russian [50,61]. Another strain used less frequently that has shown efficacy similar to that of the previous strains is the Phipps [46,66,82,83]. In a study in mice where ten vaccine strains were tested, Phipps was the one that provided the best results against the number of colony-forming units (CFU) and pathological damage in the lung after a challenge with a field strain of *Mycobacterium tuberculosis* [84]. It is unknown, however, how these strains will perform in natural challenge settings.

The stability of a vaccine is an important factor, mainly in oral vaccination. The stability of the BCG Pasteur strain was lower compared to the Danish strain under similar conditions [85]. The Danish strain was shown to be stable for 3 to 5 weeks under field conditions in a forest/grass habitat, and seven weeks in the lipid matrix at room-temperature conditions (approximately 21 °C). Under frozen laboratory conditions, the stability of the BCG vaccine was up to eight months [86]. Regarding safety, parenteral BCG vaccination did not have any unsafe reactions except local abscesses or nodules at the point of inoculation or minor adverse clinical signs [26].

For how long to vaccinate.

There are factors that affect the vaccine efficacy such as disease prevalence; exposure to infection can be higher if the prevalence is high. There is also higher risk of infections previous to vaccination, and regarding the herd dynamics, high culling rates may lead to a higher global efficacy of the vaccine [87,88].

In any vaccination strategy, it is important to establish short-, medium-, and long-term goals; thus, the duration of the use of the vaccine in an infected herd will depend on the goals established. Some goals may be very demanding, such as seeking to eliminate the disease quickly, in a period of three to five years. In this case, the use of the vaccine would probably be used for a short period of time in combination with practices such as increasing voluntary culling and some other prevention measures. In an effort to achieve a goal like this, some producers could be willing to carry out partial depopulation of the herd. If the goal is to just reduce prevalence rates to levels so low that the “test-and-slaughter” strategy is economically feasible, the period of time vaccinating would be larger—eight to ten years perhaps. However, it should be understood that in any goal, the time vaccinating could be highly variable, depending on the epidemiological status of the herd and the livestock farmers’ interest in achieving the goal. Two studies at very small scale, conducted in Germany, concluded that bTB could be eliminated from the herd in about 7 years [89,90]; however, these studies were dropped at that time, due the benefits observed with the test-and-slaughter practice.

If a vaccination program to combat bTB is to be established in an infected herd, some important considerations should be specified. For example, specify that the vaccine should be used to achieve specific goals, to eliminate the disease from the herd, and not be used indefinitely, and under the supervision and control of official authorities. If the vaccine is available and easy to obtain in an open market, there is some risk that the interest of the farmers for reaching the goal is lost and the goals jeopardized.

In any vaccination strategy used, it is advisable to consider the implementation of additional actions that contribute to the epidemiological effect of the vaccine, for example, increasing the voluntary culling of animals considering comorbidities, low production, reproductive or leg problems, and some others. The faster the infected animals leave the herd, the faster the goal will be reached.

Vaccine application route.

One of the most effective health measures for the control of infectious diseases is vaccination, and the route of application used is an important factor in the immune response and vaccine efficacy, in addition to influencing the cellular and humoral immune response [91,92]. If we consider that the main route of transmission is aerogenous, the administration of a vaccine via aerogenous means directly into the respiratory mucosa could offer a physiological and immunological advantage [93]; however, some studies show that intranasal administration results in decreased expression of TNF-α mRNA [93,94].

The most widely used inoculation routes, due to their easy application, are intramuscular (IM) and subcutaneous (SC); however, IM is not considered the best route because it induces few antigen-presenting cells [95]. It has been shown that in SC vaccination, neutrophils massively infiltrate the dermis after BCG vaccination, which is important since neutrophils have been described as the inflammatory cells that provide the first line of defense against infection in various species of mycobacteria [96,97]. Intramuscular (IM) application generally leads to a better antibody response compared to subcutaneous (SC) application [92], although this difference is not seen for some vaccines [98]. In some diseases such as HepB and rabies, a lower intradermal dose induces a response similar to that induced by a higher intramuscular dose [99]. IM application is known to induce a response mediated by T cells, while intradermal application activates a response mediated by dendritic cells, which requires lower doses of antigen [100]. In relation to the vaccination of cattle with BCG, there are several alternatives; however, the most practical for use in the field is perhaps subcutaneous. This is because apart from the easy application, it allows a slower release of the antigen at the application site, which favors a longer immune response [101].

BCG vaccination has been administered by a variety of routes, including subcutaneous, intramuscular, and mucosal (conjunctival and oral); in all cases, the vaccine has induced significant protection against experimental endobronchial challenge with wild-type *M. bovis* [85,88].

Other routes used have been oral [69,70], oral and systemic [68], intratracheal [31], nasal [102,103,104], subcutaneous [46,72], and subcutaneous and endobronchial [73]. It has been reported that BCG vaccination by intravenous, intradermal, and oral routes generated some degree of resistance to challenge with high doses of *M. bovis* in milk, something similar to that obtained when the subcutaneous route was used [105]; others showed differences in response when vaccinated subcutaneously or intravenously [26], while some reported similar results in domestic and wildlife animals by aerogenous, oral, or parenteral routes [39,46,95,106]. In conclusion, even though several alternatives of vaccine application are available, for practical reasons, the subcutaneous route is the most practical for use in the field [26,29,42,46,66].

Vaccination dose.

The dose and frequency of vaccination are important for the type of protective immune response that is established and for the duration of this protection. A lower level of protection was observed with high doses (10^9^) compared to medium and low doses in parenteral administration of BCG in cattle and deer [49]. There is generally a linear relationship between the dose of the vaccine and the level of immune response [107]; however, the age of application has an impact, and maternal antibodies in colostrum or milk can interfere with the response [108], especially when live vaccines are used. Reports from several studies indicate that doses of 10^4^ to 10^6^ CFU of BCG administered parenterally have similar levels of efficacy [26,31,46,66,72], while higher doses, 108 CFU, only obtain a good response when administered orally [69,70]. In field trials, the 2–8 × 10^5^ vaccine dose has an effect on the number of BCG colony-forming units (CFU) in calves vaccinated with the Russian BCG strain subcutaneously [43,50]. Positive results were observed also when using 3 × 10^5^ CFU [49] or 1–4 × 10^6^ [48]. According to these reports, it seems that any dose around 1 × 10^6^ UFC by the subcutaneous route should provide good protection against pathological damage in vaccinated animals in any infected herd.

One important point to consider, once the decision of using the vaccine in infected herds has been taken, is to decide the vaccination strategy to use. This strategy should be based on a previous epidemiological and herd management practices evaluation and the goals established. Among many others, the following strategies could be considered.

Differential diagnosis.

If vaccination is implemented in countries that have control programs based on “test and-slaughter”, it will be necessary to have diagnostic tests to differentiate vaccinated from infected animals (DIVA tests) [109]. Field BCG strains have been shown to express ESAT-6, CFP-10, and Rv3615c proteins, while the BCG vaccine has not, allowing these antigens to be used in an IFN-gamma test to make this differentiation [54,109]. Studies with these antigens in cattle have shown a sensitivity similar to that of the tuberculin test and a specificity of 97 to 99%. Animals infected with *M. bovis* or *M. tuberculosis* do not affect the result of the DIVA test [109]. The compatibility of BCG with a fusion reagent (DSF-F), which includes ESAT-6, CFP-10, and Rv3615c, was recently demonstrated in an intradermal DIVA test. The reactive agent showed 90% sensitivity and 95% specificity [61,110]. Although promising under experimental conditions, the performance of this DIVA test under conditions of natural challenge in a field test in different regions is still pending.

The compatibility of a DIVA test and BCG vaccination is an alternative that would especially help the free marketing of animals, regardless of the economic situation of the countries [61].

Although it is true that vaccination interferes with the tuberculin test, it is also true that there are regions with bTB and no control programs where the tuberculin test is not used. In some other regions with control programs, for different reasons, the test is rarely used; usually those regions do not move animals. Even in those situations, the vaccine could be useful. In such situations, the interference of vaccination with the tuberculin test is irrelevant and even so, vaccination could be beneficial; the evaluation of the vaccination program could be carried out by monitoring the presence of lesions of TB in carcasses and the number of seizures in slaughterhouses.

### 2.3. bTB Vaccination Strategies

#### 2.3.1. Whole-Herd Vaccination

This means that initially all animals in the herd, young and old, infected and healthy, are vaccinated. On the one hand, as already mentioned, age is not a factor that significantly influences the efficacy of the vaccine [50,51,72]. On the other hand, it has been shown that, although the vaccine does not cure the infected, it does not exacerbate the disease either [74]; that is, vaccinating infected animals will not help reduce the impact of the disease, but it will not aggravate their health status either. However, in healthy animals, it can first prevent the development of lesions, and second, reduce the rate of spread of the disease in the herd, as has been shown experimentally by reducing the number of sick animals, the number and size of lesions, as well as the bacterial load. After the program starts, a routine vaccination strategy for calves between three and five months of age should be continued. Depending on the risk of contagion that is perceived in a previous evaluation of the herd, the starting age can be reduced. This calf vaccination strategy may well be integrated into the current brucellosis vaccination program, which is one of the best-established programs in commercial herds in several countries. In this scheme, it is highly recommended to use a protein-based booster between one and two months post-vaccination in the case of adults, and two to three months before calving when starting with young; this is to ensure that when heifers enter into the production herd, the immune response is at the peak, increasing the chances of protection. It has been estimated that the duration of post-vaccination immunity is between 1 and 2 years, so revaccination is highly recommended. Studies have shown that revaccination at this time protects better than no revaccination [108].

An important factor in disease control could be the size of the herd; large herds have been associated with an increased risk of infection [111], and thus tend to have higher prevalence, which would increase the rate of exposure and, therefore, the spread of the disease through close contact. In some European countries and the north of the United States, the herd size of beef cattle can be small, 50 to 70 animals on average; in Africa and Central and South America, they can be a little larger, between 70 and 100 animals, but they do not reach the size of milk production units, which in different countries can house between 500 and 5000 or more animals. Therefore, the vaccination strategy could be different depending on the production system.

#### 2.3.2. Vaccination of Heifers

In dairy farming, with the right management, calves have low chances of becoming infected since they are separated from the mother immediately after birth, preventing infection by direct contact. Then, they are fed colostrum from healthy cows or heat-treated colostrum, and later fed milk from healthy cows or milk substitutes. In some dairy premises, calves are raised in separated farms, sometimes several kilometers apart from the production herd, where they are kept until ready for calving. At this stage, they are returned to the original farm where, immediately after calving, go into the production herd, where the risk of infection is high. It is at this stage where vaccination could play a pivotal role in preventing infection. With this background, one recommendation would be to vaccinate heifers, first with the BCG strain, and after two to three months with a cultured protein filtrate (CPF). Therefore, the vaccine should be applied two to three months before calving, just before returning to the original infected herd.

Using the CPF as a booster to the primo vaccination with BCG has been shown to improve efficacy in reducing the effect of infection considering different parameters [26,46,57,112]. With this vaccination strategy, the aim is to make heifers get into the infected herd with their immune response to the highest level, reducing the chances of infection. Studies have shown that both the BCG as well as the CPF are safe for the pregnant heifer and her fetus, and can therefore be used in pregnant animals [66]. In addition, it has been reported that vaccination of heifers has positive effects on their productive and reproductive behavior in at least the first lactation [50]. The duration of the immune response after vaccination has been estimated at between 12 and 24 months [71,74], although others under natural conditions have determined it to be shorter at 6 to 12 months [43]; in any case, this allows enough time for protein boosting.

#### 2.3.3. Calf Vaccination

Although it is considered that TB infection of dairy calves occurs in the first month of life through contaminated colostrum or milk [113], the reality is that with proper management, the chances of infection could be low; therefore, a good vaccination strategy can lead calves to reach adult life free of the disease. Thus, herds interested in vaccinating without haste, little effort, and long-term goals can choose this strategy. In this case, the objective is to prevent the disease in calves, ignoring adult animals. Calf vaccination against tuberculosis can start from five to six months of age [31,46], or younger, depending on the potential risk of infection perceived by the veterinary doctor or the person responsible for the program. One to two months after the BCG, a protein booster should be applied [46,108] with an appropriate adjuvant if preferred. This or a different boost could be applied two to three months before calving [66]. As mentioned, this calf vaccination strategy can be integrated into the current brucellosis vaccination program, which is one of the best-established programs in commercial dairy herds in several countries.

### 2.4. Additional Considerations Favorable to the Vaccine

Although there are not always scientific studies that support them, some events in nature are known to influence daily life, for example, environmental factors [114], pathogens, or animal populations’ behavior, and some others may influence the success of disease control programs in animals. In the case of vaccination against tuberculosis, some events that are expected to favor the success of the program are:The vaccine strain is of bovine origin, M. bovis; therefore, its ability to protect should be better in cattle than in humans.The natural challenge doses are surely lower than the doses used experimentally [58], so that, with vaccination, the chances of infection and consequent spread of the disease will also be lower.Although it is not known precisely, the natural rate of dissemination from infected to susceptible animals in the infected herd is low, and in tuberculin tests, the herd never reaches 100% of reactors; therefore, the vaccine has high chances of competing for the susceptible animals.The populations of dairy cattle are very dynamic; the cows’ average productive life is between three to four years of age, that is 2.5 to 3 calving seasons [115], which favors the probability of rapid elimination of animals infected with tuberculosis.Only 1% of the infected cows shed the bacillus in milk, and most of this milk goes to pasteurization, so the chances of calf infection through this means are low.Although it has not been determined in cattle, it is estimated that the rate of infection after exposure is similar to that observed in humans (≈10%) and of these, only 5% develop the disease.Finally, natural factors adverse to M. bovis in the environment, such as susceptibility to heat, sunlight, and changes in pH [112,116,117], may also contribute to a greater impact of the vaccine on the spread of the disease under natural conditions.

The alternatives of vaccination strategies in an infected herd can be many, depending on the goals set and the producer’s interest in meeting them, where a good epidemiological evaluation of the herd previous to applying the vaccine can be of great help [118,119]. It is pertinent to mention that, as occurs in many disease control programs, vaccination can have a noticeable immediate impact in infected herds with high prevalence; however, to completely eliminate the disease could be a long-term process, and that must be clearly explained to the farmers. Some of the points to consider before the implementation of a vaccination in the infected herd program are:The initial prevalence of the disease, determined by the tuberculin or the IFN-gamma tests. The prevalence of the disease may have an impact in the efficacy of the disease; high prevalence means more chances of exposure to infection.The quality of the colostrum used to feed the calves: natural or heat-treated. In addition to the immunological quality, colostrum should not pose a risk of infection to the calf; therefore, it should come from healthy cows or from a previous heat treatment [120,121].The quality of the milk used to feed the calves: natural or substitute. As with colostrum, milk for calves should come from healthy animals or powdered substitutes. Although in a low proportion, the digestive tract is a likely route of infection, especially in cases of tuberculous mastitis [121].The level of separation between the calf-rearing pens and the corrals for heifers with the corrals of the infected herd of adults. This separation must be sufficient to prevent the passage of infection through close contact or any type of fomite, such as water or food, considering that the main route of infection is respiratory and, to a lesser extent, digestive [121,122].Management practices for feed waste between pens. Using this waste to feed heifers and dry cows should be avoided. As in the previous two cases, contaminated food can be a route of infection [122].The flow of water supply: free herd first and the infected herd later [123,124].The introduction of TB-free animals as a replacement or using animals from the same herd. It has been shown that the main mechanism of disease entry into a herd is through the entry of infected animals [125].Use wisely, and with the purpose of eliminating TB, the decision of voluntary and involuntary culling.

### 2.5. Vaccination and Biosecurity

Few vaccines by themselves are capable of preventing the presence of a disease in a population, even when the levels of efficacy are high. It is known that biosecurity measures always contribute to the success of disease control programs, and in a TB vaccination program, it should not be different; the minimum number of measures to be implemented in the vaccinated herd are:Separate the tuberculin-reacting animals and handle them separately.Make real separation between calves’ pens and adults’ settings.Use only replacements from the same herd or TB-free if they come from another herd.Feed calves with colostrum and milk from healthy cows.Milk healthy cows first.Avoid giving waste feed from the infected herd to calves, heifers, or dry cows.Organize pens and direct the flow of water from the free to infected pens.Prevent the entrance of out-of-the-farm vehicles to the animals’ area.Take the necessary provisions with cattle that move to fairs and exhibitions to avoid contagion.

### 2.6. Factors That May Affect Vaccine Efficacy

It is important to remember that there are factors that affect the effectiveness of a vaccine. As is known, the BCG is a live vaccine, so poor management can represent a risk for its viability and efficacy. Therefore, it is important to follow the provider directions at any stage. Some of the known factors that can affect the effectiveness of the bovine tuberculosis vaccine are:The genetic variability of individuals: some vaccinated animals do not generate an appropriate immune response and may present visible TB lesions.The vaccine strain used: some BCG strains have been over-attenuated.The dose administered: in the case of bTB, high doses are less effective than medium or low doses [31].The vaccine protocol, some more successful than others: the use of boosting with proteins has shown better efficacy [26].The nutritional and physiological state of the animal: malnourished and pregnant animals may not develop an appropriate immune response [126].Pre-immunization with environmental mycobacteria: this can reduce the effectiveness of the BCG vaccine because the animal may have memory cells that are activated when detecting the vaccine antigen [26].The consumption of infected milk contaminated with M. bovis or with environmental mycobacteria [121].The presence of concurrent diseases.The indiscriminate use of medications (corticosteroids).

## 3. Conclusions

Cattle tuberculosis remains a costly livestock problem and public health risk in many parts of the world. The “test and kill” strategy is not feasible in many regions due to the cost of culling animals without compensation and the expense of buying replacements to maintain production standards. Research works with stringent experiments, using high challenge doses compared to that expected under natural conditions, have shown that vaccination with the BCG strain, alone or in combination with protein preparations, has the potential to reduce the impact of the disease in vaccinated animals, reducing the rate of infection, the magnitude and number of visible lesions, as well as the bacillary count. Therefore, the incorporation of the vaccine in current control programs would undoubtedly help reduce the impact of this costly disease both in domestic animals and in free-living animals. A greater number of field trials under natural conditions of challenge may confirm or contradict this assertion.

## Figures and Tables

**Table 1 animals-12-03377-t001:** Efficacy of the vaccine BCG against tuberculosis in field trails in cattle.

Trial	Country	BCG Strain	Vaccine Dose	Vaccine Route	Age at Vaccination	Challenge (Prevalence = Proportion of Reactors to Immunological Tests)	Efficacy Based on	Efficacy	Reference
1	Mexico	Tokyo	10^6^	* SC	1–2 weeks	Infected herd (40% prevalence)	Proportion of positive to three immunological tests	59.4%	2010 [41]
2	Ethiopia	Danish	10^6^	* SC	2 weeks	100% prevalence	Proportion with lesions	56–61%	2010 [42]
3	Ethiopia	Danish	1–4 × 10^6^	* SC	2 weeks	100% prevalence	Proportion with lesions	23–28%	2018 [48]
4	New Zealand	Danish	10^8^	Oral	6–30 months	5–10% prevalence	Pathology score Proportion with lesions	67.4%	2018 [49]
5	Chile	Russia	2–8 × 10^5^	* SC	11 months	24% prevalence		66.5%	2022 [50]
6	Chile	Russia	2–5 × 10^5^	* SC	40 days	Seven herds included with 15–75% prevalence	** IFN release + antigens *** ESAT-6, **** CFP-10 and Rv3615c	22.4%	2022 [43]

Modified from [26]. * Subcutaneous. ** Interferon. *** Early Secreted Antigenic Target 6 kDa. **** Purified Protein Filtrate.

**Table 2 animals-12-03377-t002:** Experiments showing the efficacy of the BCG vaccine in preventing visible lesions in vaccinated animals vs. controls.

Trial	Country	BCG Strain	Dose	Vaccination Route	Significant Vaccine Protective Effect for Pathological Damage, Vaccinated vs. Controls	Reference
1	New Zealand	Pasteur Attenuated *M. bovis* WAg500 Attenuated *M. bovis* WA501	1 × 10^5^ 1 × 10^6^ 2 × 10^6^	SC	Yes Yes Yes	2002 [67]
2	New Zealand	Pasteur	10^6^ 8 ** h of birth 10^6^ 8 h of birth 10^6^ 6 w of birth	SC	Yes Yes Yes	2003 [29]
3	United Kingdom	Pasteur	10^6^	SC	Yes	2005 [51]
4	New Zealand	Danish Pasteur	10^6^	SC	Yes Yes	2005 [57]
5	New Zealand	Pasteur	10^6^ SC 10^9^ 10^6^ SC + 10^9^	SC Oral/SC Oral/Oral	Yes Yes Yes Yes No significant better effect by using both routes	2008 [68]
6	Mexico	Tokyo	1 × 10^6^	SC	Yes	2010 [41]
7	New Zealand	UK	2 × 10^7^ 2 × 10^7^ + CFP CFP + emulsigen 10^8^ 10^6^	Oral Nasal SC Oral SC	Yes No No No Yes Yes	2011 [69]
8	United Kingdom	Pasteur Danish	2 × 10^6^ 2 × 10^6^	SC SC	Yes Yes	2011 [30]
9	New Zealand	Danish	10^8^ Oral 10^7^ Oral 10^6^ 10^6^ SC	Oral SC	10^8^ Oral, yes 10^7^ Oral, no 10^6^ Oral, no 10^6^ SC, yes	2011 [70]
10	United Kingdom	Danish	1 × 10^6^ 4 × 10^6^	SC	At 12 months, yes At 24 months, no	2012 [71]
11	New Zealand	Danish	1 × 10^5^ to 4 × 10^5^ or 1 × 10^6^ to 4 × 10^6^	SC	Yes	2013 [72]
12	United Kingdom	Danish	10^6^ 10^6^ 5 × 10^5^ BCG + 5 × 10^5^ BCG SSI 10^6^ BCG SSI + 2 × 10^9^ *** Ad85A	SC EB SC + EB * simultaneous SC + EB * simultaneous	No No Yes Yes	2015 [73]
13	Mexico	Phipps	1 × 10^6^	SC	Yes	2013 [46]
14	New Zealand	Danish	10^5^ with no revaccination 10^5^ with revaccination	SC	No Yes	2013 [72]
15	New Zealand	Danish	1 × 10^6^	SC	No	2016 [74]

* Endobronchial. ** Hours. *** Recombinant, replication-defective human type 5 adenovirus expressing *M.tb* Ag85A.

## Data Availability

The data presented in this study are available on request from the corresponding author.

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
