# Peer review of "Vaccination Strategies in a Potential Use of the Vaccine against Bovine Tuberculosis in Infected Herds"

_animals, 2022, doi:10.3390/ani12233377_

Round 1

Reviewer 1 Report

Dear authors

In my humble opinion, a review of a more recent bibliography would benefit the results.

The bibliography is very extensive but some outdated. For example, the trend of tuberculosis cases in the European Union is assessed by the 2007 report, when there are more recent reports. The 2017 one presents much more reliable data “The European Union summary report on trends and sources of zoonoses, zoonotic agents and food-borne outbreaks in 2017”.

The reality in Africa is very heterogeneous and the bibliography is mainly based on Ethiopia. It would be interesting to draw on the experience of South Africa, which has a worrying reality and within the objectives of your work: “Challenges for controlling bovine tuberculosis in South Africa”, published 2020 would be beneficial for the discussion. 

Your work constitutes a good review of the literature and provides a basis for programming future pilot studies.

Author Response

Dear reviewer thank you very much for your comments. 

The suggested references were check and incorporated in the paragraph related to the situation of bTB in Africa (lines 47-53).

We appreciate this kind comment.

Reviewer 2 Report

Dear Authors,

This review's significant contribution is the questions anyone can ask about public health worldwide and what everyone can do to contribute to the "ONE HEALTH". The Review shows important facts about tuberculosis vaccination, shows its advantages, and some limitations. It brings a profound discussion about the importance to vaccinate and puts arguments in pros and cons to make the reader think about it. I think it should be of great interest to the scientific community concerned about public health.

Author Response

Dear reviewer thank you very much for your comments. 

Reviewer 3 Report

This paper is a brave attempt to review and consolidate the evidence for the impacts that BCG vaccination may have at the animal and herd level through the lens of making recommendations for strategies for the use of the vaccine in infected herds. The paper is structured into two halves, the first of which takes the form of a non-systematic review of the (messy) history and literature of experimental and (rare) attempts at field evaluation. The review is broadly well written and comprehensive missing only a few edge studies (early field trials of BCG in England and Africa). However, there is little attempt made to deal with the the messiness within the literature in terms of the lack of consistency in how vaccine efficacy has been defined in past studies, along with the considerable uncertainty surrounding the epidemiological context (studies been carried out globally over a period of > 100 years) and composition of individual vaccines and studies. BCG is a live product which has been continuously cultured by different manufacturers leading to considerably differences in potency in the products used by different authors. While the first section is closely referenced it is largely limited to quoting results (or the massive range of results) from past studies without placing them into context. This context is essential in terms of weighing the likely quality and relevance of studies, especially given the large number of studies with very small sample sizes or limited follow-up time (Given which we might expect to see large variation in estimated effects sizes even in the absence of the many other sources of variation between studies discussed above and in the paper itself). 

The second half (section 2.3.3 onwards) presents a series of recommendations for potential strategies for the use of vaccination as part of a control program are discussed. In contrast to the preceding review of the literature almost no claims in the section are referenced at all making it much weaker and vaguer. Some of the statements in this section, while plausible - and certainly in line with established dogma - would be exceptionally difficult to evidence. This section could be considerably strengthened - and be much more useful - if there was a careful linking to the evidence (or rationale) for each suggestion. 

However, a larger issues is that the review does not address at all what the policy objectives (or reasonable expectations) for the use of vaccination should be. Some of these questions are set up in terms of what the objectives might be -  e.g. reducing prevalence or clearing herds from infection - but there is no discussion of the evidence for why this would be beneficial. Evidence on the impact of bTB on productivity is sparse and mostly comes from studies from Germany in the 1950s which are of questionable relevance for modern production systems in diverse cattle and dairy production systems. Likewise the motivations for the use of vaccination will be very different in countries which do not currently have extensive test-and-control programs. While a relatively poor vaccine may have considerably benefits in terms of reducing the burden of disease in unmanaged herds - it may only add costs for farmers (or government) when added as a supplement to existing test and slaughter. The challenges of deploying vaccination in managed herds is further increased due to the interference of BCG with the action of the tuberculin skin test. Even for countries with no current control program, international trade depends on the tuberculin skin test, meaning that without a test to differentiate infected from vaccinated animals (DIVA), farmers may be severely limited in their ability to trade. The lack of discussion of the prospects for a DIVA test (or the limitation that vaccinated animals would not be able to be traded) is therefore a huge omission - which I would argue is central to the prospects for any deployment of BCG in cattle anywhere.

Despite the large body of work on the potential use of BCG in cattle, very little work or discussion has considered what the business case for a vaccine should be. As above, this is perhaps a much bigger topic than the authors wish to tackle - particularly as the details depend critically on the epidemiological, social and political context which will vary hugely in different target markets. In human epidemiology no vaccine would be introduced without a clear business case which outlines the (likely) costs and benefits of interventions. These considerations are all the more important if farmers are expected to bear the costs of vaccination as they should have a reasonable expectation of what the vaccine is likely to benefit them. The scope of addressing this could be made more tractable by refocusing the topic of the review to unmanaged herds - or even just to Mexico as this may allow the authors to make more concrete - and valuable synthesis of what the current knowledge around BCG efficacy suggests for what the business case of BCG may be in their country - which will be very different to other contexts around the world.

Minor comments

Line 52. I believe OiE has now rebranded to WOAH.

Line 71. Suggest removing "and overcrowding favors the dissemination" while higher prevalence is evidenced in larger herds the mechanism is not clear (see discussion below).

Line 92. Can you clarify what conclusion is doubtful? This meta-analysis estimated efficacy from (largely experimental challenge studies) that had a consistent measure of vaccine efficacy (sterilising immunity to infection/absence of lesions). Given this definition of efficacy the relatively low efficacy is actually encouraging as the primary claim for BCG in cattle is to reduce the extent of lesions. We would expect efficacy under natural transmission conditions to be higher (indeed the authors discuss this very issue at length in the meta-analysis).

Line 116. There is a conflation here between willingness to pay and the cost of the vaccine. While the willingness to pay in the UK could be as high as £17, BCG can be acquired on the open market for much cheaper...

Line 106. I certainly agree that BCG is likely to reduce the prevalence of disease and incidence as measured by the rate of new infections. However, as incidence is typically measured in terms of test positives it should be acknowledged that (official/observed) incidence could actually be expected to increase in herds that vaccinate even while prevalence is reduced. The extent of this effect (and duration) will depend on vaccine efficacy and the characteristics of any replacement DIVA test - all the more reason this really needs to be discussed in the review.

Line 130. Reference 59 does not discuss or estimate in anyway the impact of research studies on productivity within farms, only quantifying a difference in herd prevalence between herds they had assessed to have "good" or "bad" management. Not clear to me what the 10% figure quoted here refers to, what "each management" is or how production is measured?

Lines 131-146. As above, given the differences in the endpoint for measuring efficacy I question whether it is useful to summarise the estimates within a single range given than each study is potentially measuring a different quantity. Mapping or tabulating the differences in endpoint (and sample sizes) would help to clarify the origins of this variability.

Line 154.  "to identify the levels of these proteins in the vaccinated animals to predict protection and pass to second term the tuberculin tests." What is the second term and how does identifying correlates of protection relate to the tuberculin test? Is the point here that it may help develop a DIVA - there are candidate DIVAs published in several studies in the literature now (and not discussed in this review).

Line 156. As above, all of these technical questions as to how to use the vaccine bests come after what the objective/justification for the use of vaccination is: 

To prevent disease/harm in animals?

To reduce the risk of zoonotic transmission?

To increase productivity?

Discussion for the evidence (or lack of) with respect to rates and routes of zoonotic transmission and the impact of bTB on productivity would seem to me to be central to any argument for vaccination.

Line 200. I assume the authors mean "eliminate" - eradication refers specifically to elimination of a pathogen globally.

Line 200. Is there any prospect that BCG could plausibly eliminate disease from a herd within less than five years? Again with no clear discussion of policy objectives and what is reasonable to expect given the efficacy of BCG and uncertain/variable rates of transmission this figure could easily be confused with a prediction/realistic expectation rather than a more vague goal.

Line 212. The two cited studies suggesting that vaccination could eliminate bTB from herds within 7-8 years come from Germany in the 1950s. These studies are impossible to find online (and only in German) so would be helpful to put this figure into more context (or at least acknowledge this has only been attempted in very small scale studies which were abandoned at the time due the greater efficiency of test-and-slaughter)

Line 221. Do the authors mean "aerosol" transmission rather than "erogenous"?!

Line 304. Transmission rates of bovine TB are often assumed to be density dependent (strictly herd size dependent), however the evidence for this is extremely weak amounting to a single figure in the textbook by John Francis in 1947. Large herds are not necessarily overcrowded as farmers may choose to manage them in smaller units. What is clear from many studies, globally, is that larger herds tend to have a higher prevalence within-herds and a higher risk of being affected by TB. The mechanisms for these observed risks could include higher rates of cattle-to-cattle transmission but could also reflect a higher risk of exposure from the environment, risks from cattle movements or even differences in susceptibility surrounding the management of the herd. Larger herds may well have a higher rate of transmission which would make control by vaccination more difficult. I would suggest that the evidence for this lies in the empirical association of prevalence with herd size and the evidence for this (such as it is) should be cited rather than continuing to perpetuate dogma.

Line 343: extra "about"

Line 357: This statement is supposition - what is the evidence for this? 

Line 359: While I agree that vaccination might be expected to more effective under natural transmission than challenge, given the lack of evidence for even the mode of transmission of bTB (see decades of speculation with respect to aerosol, feaco-oral, environmental, pseudo-vertical etc.) how can we say that the dose will be lower?

Line 362: With estimates of the sensitivity of the tuberculin test falling as low as 50% in some contexts then reactor rates of as low as 50% may well map to a true prevalence of close to 100%.

Line 366: This point is also a strong argument against the effectiveness of vaccination as vaccinated (protected) animals are also being removed rapidly from herds limiting the potential for herd immunity effects (even for an efficacious vaccine)

Line 369: This point could also be used to argue against the use of vaccination to reduced risk of zoonotic transmission.

Line 371: This point is so critical to the justification for all control strategies for bTB, and as the authors state here there is no published empirical evidence that speaks to the relationship between test positivity and disease. By "estimate" do the authors mean "guess"? If this estimate is correct then the vast majority of cattle killed in TB testing programs are healthy animals. Such a bold claim surely requires some evidence or further context?

Line 374. Cannot see how this point is of any relevance to the effectiveness of a vaccine as efficacy is measured with respect the relative risk of acquiring disease relative to unvaccinated individuals in the same population. The absolute risk should not matter (expect with respect to the potential of dose-dependent effects as discussed above.)

Line 374: As with the above list section, I would not disagree with the general sentiment of each of these points, however the lack of direct evidencing or specific advice (e.g. qualitative suggestion to "Use wisely") does not for me provide a valuable contribution to the key question of what the business case for vaccination should be (or how we can come to that conclusion).

Line 383: Again "eliminate" rather than "eradicate" if your scope is at the farm level only.

Line 446: Although I am a huge proponent of vaccines and whole-heartedly agree that BCG can reduce the prevalence of disease within herds, I very much doubt that it will help reduce the impact of the disease in all epidemiological contexts around the world. In particular as discussed above the business case in countries with extensive test-and-slaughter programs is fraught with difficulties and complications. Impacts of vaccination must also include the economic, wider animal and public health costs and benefits of the intervention which are not addressed at all in this review.

Author Response

REVIWER 3

Reviewer comment (rc). This paper is a brave attempt to review and consolidate the evidence for the impacts that BCG vaccination may have at the animal and herd level through the lens of making recommendations for strategies for the use of the vaccine in infected herds. The paper is structured into two halves, the first of which takes the form of a non-systematic review of the (messy) history and literature of experimental and (rare) attempts at field evaluation. The review is broadly well written and comprehensive missing only a few edge studies (early field trials of BCG in England and Africa). However, there is little attempt made to deal with the the messiness within the literature in terms of the lack of consistency in how vaccine efficacy has been defined in past studies, along with the considerable uncertainty surrounding the epidemiological context (studies been carried out globally over a period of > 100 years) and composition of individual vaccines and studies. BCG is a live product which has been continuously cultured by different manufacturers leading to considerably differences in potency in the products used by different authors. While the first section is closely referenced it is largely limited to quoting results (or the massive range of results) from past studies without placing them into context. This context is essential in terms of weighing the likely quality and relevance of studies, especially given the large number of studies with very small sample sizes or limited follow-up time (Given which we might expect to see large variation in estimated effects sizes even in the absence of the many other sources of variation between studies discussed above and in the paper itself). 

Authors response. We really appreciate the time reviewer 3 expend in reviewing our manuscript, we found his comments very appropriated and important to improve the quality of our paper. He could be an excellent co-author if he or she was interested, his/her knowledge and experience in this paper topic would be greatly appreciated. However, some of his comments are a little bit difficult to understand, for example, we don´t understand what exactly the reviewer means by “… place the different experiments into context to weigh the relevance of the studies”. For what we understood, totally agree with the comment, there are quite some differences in the large number of experiments about the efficacy of the BCG. In fact, this is one of the reasons why a meta-analysis including all the experiments has not been performed. To satisfy the reviewer concern we add a paragraph stressing this issue on lines 88-104.  

The second half (section 2.3.3 onwards) presents a series of recommendations for potential strategies for the use of vaccination as part of a control program are discussed. In contrast to the preceding review of the literature almost no claims in the section are referenced at all making it much weaker and vaguer. Some of the statements in this section, while plausible - and certainly in line with established dogma - would be exceptionally difficult to evidence. This section could be considerably strengthened - and be much more useful - if there was a careful linking to the evidence (or rationale) for each suggestion. 

Authors response. Again, we agree with the reviewer’s concern. Most of the recommendations made were referenced, when possible, some others are common sense and based in the authors and colleagues experience.

However, a larger issue is that the review does not address at all what the policy objectives (or reasonable expectations) for the use of vaccination should be. Some of these questions are set up in terms of what the objectives might be -  e.g. reducing prevalence or clearing herds from infection - but there is no discussion of the evidence for why this would be beneficial. Evidence on the impact of bTB on productivity is sparse and mostly comes from studies from Germany in the 1950s which are of questionable relevance for modern production systems in diverse cattle and dairy production systems. Likewise the motivations for the use of vaccination will be very different in countries which do not currently have extensive test-and-control programs. While a relatively poor vaccine may have considerably benefits in terms of reducing the burden of disease in unmanaged herds - it may only add costs for farmers (or government) when added as a supplement to existing test and slaughter. The challenges of deploying vaccination in managed herds is further increased due to the interference of BCG with the action of the tuberculin skin test. Even for countries with no current control program, international trade depends on the tuberculin skin test, meaning that without a test to differentiate infected from vaccinated animals (DIVA), farmers may be severely limited in their ability to trade. The lack of discussion of the prospects for a DIVA test (or the limitation that vaccinated animals would not be able to be traded) is therefore a huge omission - which I would argue is central to the prospects for any deployment of BCG in cattle anywhere.

Authors response.The reviewer calls our attention to clarify the objectives of the vaccination program. To satisfy this request we added a paragraph on lines 78 to 104. However, in our opinion, the objectives could be different for the different countries, and even in different regions in the same country. To support our point, we use the example of the bTB situation in Mexico, nevertheless, based on the status of the disease and the many factors around a possible control program, the objectives can vary. We explained this in the paragraph added.

In relation to the DIVA test, we agree, it is quite important for the purpose of this paper, therefore, we added a completely new paragraph to discuss the importance of this kind of tests in a potential bTB vaccination program, this paragraph is on lines 377-399. 

Despite the large body of work on the potential use of BCG in cattle, very little work or discussion has considered what the business case for a vaccine should be. As above, this is perhaps a much bigger topic than the authors wish to tackle - particularly as the details depend critically on the epidemiological, social and political context which will vary hugely in different target markets. In human epidemiology no vaccine would be introduced without a clear business case which outlines the (likely) costs and benefits of interventions. These considerations are all the more important if farmers are expected to bear the costs of vaccination as they should have a reasonable expectation of what the vaccine is likely to benefit them. The scope of addressing this could be made more tractable by refocusing the topic of the review to unmanaged herds - or even just to Mexico as this may allow the authors to make more concrete - and valuable synthesis of what the current knowledge around BCG efficacy suggests for what the business case of BCG may be in their country - which will be very different to other contexts around the world.

Authors response. We agree with the reviewer in the sense that the cost per doses of the vaccine could be lower in an open market, in the paragraph in lines 173-184 we provide some information about what the estimates of the cost could be and what the producers, at least in Mexico are willing to pay, we added some more sentences (lines 173-184) to explain that the price for the farmer could vary for the different countries according to the country’s situation.

Minor comments 

Line 52. I believe OiE has now rebranded to WOAH.

Authors response. The correction was made.

Line 71. Suggest removing "and overcrowding favors the dissemination" while higher prevalence is evidenced in larger herds the mechanism is not clear (see discussion below).

Authors response. The phrase was deleted as suggested.

Line 92. Can you clarify what conclusion is doubtful? This meta-analysis estimated efficacy from (largely experimental challenge studies) that had a consistent measure of vaccine efficacy (sterilising immunity to infection/absence of lesions). Given this definition of efficacy the relatively low efficacy is actually encouraging as the primary claim for BCG in cattle is to reduce the extent of lesions. We would expect efficacy under natural transmission conditions to be higher (indeed the authors discuss this very issue at length in the meta-analysis).

Authors response. We eliminated the word “doubtful” from the sentence. However, we still disagree with this meta-analysis conclusion, although the authors reduce the number of papers in the analysis to make it more uniform, some differences in doses, BCG strains, etc., persist. We disagree with the 25% reported.

Line 116. There is a conflation here between willingness to pay and the cost of the vaccine. While the willingness to pay in the UK could be as high as £17, BCG can be acquired on the open market for much cheaper...

Authors response. We totally agree, the cost of a drug is related to “how much it is expended in its production and many doses are sold in the open market”; however, it is always good to know what the consumer is willing to pay to make a balance and decide whether or not to go ahead with the production project. To clarify this point a short paragraph was added in lines 173-184.

Line 106. I certainly agree that BCG is likely to reduce the prevalence of disease and incidence as measured by the rate of new infections. However, as incidence is typically measured in terms of test positives it should be acknowledged that (official/observed) incidence could actually be expected to increase in herds that vaccinate even while prevalence is reduced. The extent of this effect (and duration) will depend on vaccine efficacy and the characteristics of any replacement DIVA test - all the more reason this really needs to be discussed in the review.

Authors response. We agree. In the context of the paragraph, the statement “reduce the incidence and as a consequence, the prevalence” refers to the presence of the disease, as explained in the text, and no to the response to the tuberculin test.

Line 130. Reference 59??? does not discuss or estimate in anyway the impact of research studies on productivity within farms, only quantifying a difference in herd prevalence between herds they had assessed to have "good" or "bad" management. Not clear to me what the 10% figure quoted here refers to, what "each management" is or how production is measured?

Authors response. We deleted the 10% figure from the text since we couldn´t find the reference. However, in our experience working with dairy farmers they always complain about the reduction in milk production every time we handle the cows.

Lines 131-146. As above, given the differences in the endpoint for measuring efficacy I question whether it is useful to summarise the estimates within a single range given than each study is potentially measuring a different quantity. Mapping or tabulating the differences in endpoint (and sample sizes) would help to clarify the origins of this variability.

Line 154.  "To identify the levels of these proteins in the vaccinated animals to predict protection and pass to second term the tuberculin tests." What is the second term and how does identifying correlates of protection relate to the tuberculin test? Is the point here that it may help develop a DIVA - there are candidate DIVAs published in several studies in the literature now (and not discussed in this review).

Authors response. We agree, a DIVA test is crucial in a vaccination program, it was missing in our manuscript, it was a big mistake, therefore, we included a complete paragraph about this test named “Differential diagnosis”, lines 377-399.

Line 156. As above, all of these technical questions as to how to use the vaccine bests come after what the objective/justification for the use of vaccination is: 

To prevent disease/harm in animals?

To reduce the risk of zoonotic transmission?

To increase productivity?

Authors response. To clarify this issue, we included a couple of paragraphs in lines 78-104. We consider this point important; we try to explained that the objectives can be different depending on many factors in different parts of the world.

Discussion for the evidence (or lack of) with respect to rates and routes of zoonotic transmission and the impact of bTB on productivity would seem to me to be central to any argument for vaccination.

Line 200. I assume the authors mean "eliminate" - eradication refers specifically to elimination of a pathogen globally.

Authors response.To satisfy this concern, although we didn´t include a complete paragraph related to this issue, we did, however, included some data about economic losses (lines 40-46) and the risk to public health (lines 47-48).

Line 200. Is there any prospect that BCG could plausibly eliminate disease from a herd within less than five years? Again, with no clear discussion of policy objectives and what is reasonable to expect given the efficacy of BCG and uncertain/variable rates of transmission this figure could easily be confused with a prediction/realistic expectation rather than a more vague goal.

Authors response. This issue is related to a concern previously explained, the objectives of a vaccination program. To discuss this point two paragraphs were added in lines 78-104.

Line 212. The two cited studies suggesting that vaccination could eliminate bTB from herds within 7-8 years come from Germany in the 1950s. These studies are impossible to find online (and only in German) so would be helpful to put this figure into more context (or at least acknowledge this has only been attempted in very small scale studies which were abandoned at the time due the greater efficiency of test-and-slaughter).

Authors response. This was addressed on lines 292-295.

Line 221. Do the authors mean "aerosol" transmission rather than "erogenous"?!

Authors response. Aerosol was changed for “aerogenous” which is more appropriate in the sentence.

Line 304. Transmission rates of bovine TB are often assumed to be density dependent (strictly herd size dependent), however the evidence for this is extremely weak amounting to a single figure in the textbook by John Francis in 1947. Large herds are not necessarily overcrowded as farmers may choose to manage them in smaller units. What is clear from many studies, globally, is that larger herds tend to have a higher prevalence within-herds and a higher risk of being affected by TB. The mechanisms for these observed risks could include higher rates of cattle-to-cattle transmission but could also reflect a higher risk of exposure from the environment, risks from cattle movements or even differences in susceptibility surrounding the management of the herd. Larger herds may well have a higher rate of transmission which would make control by vaccination more difficult. I would suggest that the evidence for this lies in the empirical association of prevalence with herd size and the evidence for this (such as it is) should be cited rather than continuing to perpetuate dogma.

Authors response. We agree with the reviewer and the paragraph was modified and referenced.

Line 343: extra "about"

Authors response. The extra “about” was eliminated.

Line 357: This statement is supposition - what is the evidence for this? 

Line 359: While I agree that vaccination might be expected to be more effective under natural transmission than challenge, given the lack of evidence for even the mode of transmission of bTB (see decades of speculation with respect to aerosol, feaco-oral, environmental, pseudo-vertical etc.) how can we say that the dose will be lower?

Authors response. We agree, there is not evidence; however, this is common sense.

Line 362: With estimates of the sensitivity of the tuberculin test falling as low as 50% in some contexts then reactor rates of as low as 50% may well map to a true prevalence of close to 100%.

Authors response. We agree, in theory this argument seems reasonable, however, in our experience, in cases of total herd depopulation with 30-40% tuberculin test reactors, only in one case 60% had lesions suggestive of bTB and were culture-positive.

Line 366: This point is also a strong argument against the effectiveness of vaccination as vaccinated (protected) animals are also being removed rapidly from herds limiting the potential for herd immunity effects (even for an efficacious vaccine).

Authors response. Although this could be true if new animals entering the vaccinated herd are no vaccinated, but if the replacements are vaccinated, the “herd immunity” status can be accomplished.

Line 369: This point could also be used to argue against the use of vaccination to reduced risk of zoonotic transmission.

Authors response. Sorry, we don´t understand how or why this is against vaccination to reduced zoonotic disease. Reducing the prevalence by any means necessarily reduces the risk for people, since the chances of having cows with udder mastitis is going to be lower.

Line 371: This point is so critical to the justification for all control strategies for bTB, and as the authors state here there is no published empirical evidence that speaks to the relationship between test positivity and disease. By "estimate" do the authors mean "guess"? If this estimate is correct then the vast majority of cattle killed in TB testing programs are healthy animals. Such a bold claim surely requires some evidence or further context?

Authors response. Yes, “estimate” means “guess”, unfortunately there is no evidence to support this argument.

Line 374. Cannot see how this point is of any relevance to the effectiveness of a vaccine as efficacy is measured with respect the relative risk of acquiring disease relative to unvaccinated individuals in the same population. The absolute risk should not matter (expect with respect to the potential of dose-dependent effects as discussed above.)

Authors response. The point in line 374 relates to environmental factors that affect in any way to the M. bovis bacillus, therefore, globally, factors that affect the bacillus in an indirect way would “help” to the vaccine to reduce prevalence in the infected herds.

Line 374: As with the above list section, I would not disagree with the general sentiment of each of these points, however the lack of direct evidencing or specific advice (e.g. qualitative suggestion to "Use wisely") does not for me provide a valuable contribution to the key question of what the business case for vaccination should be (or how we can come to that conclusion).

Authors response. Yes, we agree with the reviewer comment. Unfortunately, no scientific evidence to support the point is available.

Line 383: Again "eliminate" rather than "eradicate" if your scope is at the farm level only.

Authors response. The word “eradicate” was changed for “eliminate”.

Line 446: Although I am a huge proponent of vaccines and whole-heartedly agree that BCG can reduce the prevalence of disease within herds, I very much doubt that it will help reduce the impact of the disease in all epidemiological contexts around the world. In particular, as discussed above the business case in countries with extensive test-and-slaughter programs is fraught with difficulties and complications. Impacts of vaccination must also include the economic, wider animal and public health costs and benefits of the intervention which are not addressed at all in this review.

Authors response. Yes, we agree. This point is so important that a specific paper needs to be written just to discuss the context of the benefits of using the vaccine against TB in cattle in different contexts around the world. This would need the participation of co-authors from different countries, since every country context could be different: prevalence, resources to support the program, veterinary services including laboratories, etc.

Submission Date

07 October 2022

Date of this review

24 Oct 2022 15:24:16

Round 2

Reviewer 3 Report

Many thanks to the authors for addressing my major comments and concerns and providing valuable additional context and a clearer delineation between evidence and opinion.

With respects to the authors doubts about the low estimated efficacy from the recent meta-analysis of BCG efficacy I would like to clarify that I have no issue with them raising doubt about the figure, rather that they should be more specific in saying that they believe the true efficacy is higher (which is clearer now in the revised manuscript). The baseline estimate from the recent meta-anaylsis is - in my view - less important and interesting than the light that the effort shines on the lack of high quality, comparable estimates of efficacy despite over a century of effort.

I think the review has been greatly strengthened by these efforts. I would still recommend a thorough proof-reading of the manuscript for spelling, grammar and typographical errors before publication (some of which are potentially misleading e.g. powers of 10 throughout are missing superscripts).